# Bacterial Diversity Analysis and Screening for ACC Deaminase-Producing Strains in Moss-Covered Soil at Different Altitudes in Tianshan Mountains—A Case Study of Glacier No. 1

**DOI:** 10.3390/microorganisms11061521

**Published:** 2023-06-07

**Authors:** Yanlei Shi, Ye Yuan, Yingying Feng, Yinghao Zhang, Yonghong Fan

**Affiliations:** Xinjiang Key Laboratory of Biological Resources and Genetic Engineering, College of Life Science & Technology, Xinjiang University, Urumqi 830046, China

**Keywords:** global warming, primary succession, high-throughput sequencing, ACC deaminase

## Abstract

The elevation of the snowline of the No. 1 Glacier in the Tianshan Mountains is increasing due to global warming, which has created favorable conditions for moss invasion and offers an opportunity to investigate the synergistic effects of incipient succession by mosses, plants, and soils. In this study, the concept of altitude distance was used instead of succession time. To investigate the changes of bacterial-community diversity in moss-covered soils during glacial degeneration, the relationship between bacterial community structure and environmental factors was analyzed and valuable microorganisms in moss-covered soils were explored. To do so, the determination of soil physicochemical properties, high-throughput sequencing, the screening of ACC-deaminase-producing bacteria, and the determination of ACC-deaminase activity of strains were performed on five moss-covered soils at different elevations. The results showed that the soil total potassium content, soil available phosphorus content, soil available potassium content, and soil organic-matter content of the AY3550 sample belt were significantly different compared with those of other sample belts (*p* < 0.05). Secondly, there was a significant difference (*p* < 0.05) in the ACE index or Chao1 index between the moss-covered-soil AY3550 sample-belt and the AY3750 sample-belt bacterial communities as the succession progressed. The results of PCA analysis, RDA analysis, and cluster analysis at the genus level showed that the community structure of the AY3550 sample belt and the other four sample belts differed greatly and could be divided into two successional stages. The enzyme activities of the 33 ACC-deaminase-producing bacteria isolated and purified from moss-covered soil at different altitudes ranged from 0.067 to 4.7375 U/mg, with strains DY1–3, DY1–4, and EY2–5 having the highest enzyme activities. All three strains were identified as *Pseudomonas* by morphology, physiology, biochemistry, and molecular biology. This study provides a basis for the changes in moss-covered soil microhabitats during glacial degradation under the synergistic effects of moss, soil, and microbial communities, as well as a theoretical basis for the excavation of valuable microorganisms under glacial moss-covered soils.

## 1. Introduction

Numerous studies at home and abroad have found that glaciers have been undergoing a worldwide retreat. Since the 1980s, global warming has been increasing, which has significantly increased the melting rate of glaciers worldwide, resulting in the regression and accelerated ablation of glaciers and snow and even the possibility of their disappearance [1,2,3,4,5,6,7,8,9]. The massive retreat of glaciers has caused rocks covered by ice and snow layers to be released to form oligotrophic, vegetation-free, native bare land. The exposed sediments from glacier melting provide new habitats for microorganisms and excellent opportunities for the study of community construction and the diversity of extreme environmental microorganisms at the glacier front during primary succession [10,11]. Since the 21st century, a large number of researchers have realized that glaciers are a unique ecosystem wherein microorganisms from all domains of life can colonize, creating a glacial ecosystem dominated by both cold-loving and cold-tolerant microorganisms [12,13,14]. Tianshan’s No. 1 Glacier has experienced greatly decreased glacier area, decreased thickness, and rapidly increased snow-line altitude in the last 25 years, and its ablation rate is predicted to continue to increase in the future based on the current trend of global warming [15]. Tianshan’s No. 1 Glacier’s front freeze–thaw zone is in the alpine-glacier zone, which has the climatic characteristics of thin air, sudden temperature changes, low partial pressure of oxygen, strong ultraviolet radiation, and strong wind [16].

In the process of glacier dissolution and degradation, glacier ecosystems undergo succession, and bryophytes are the most seen plants here, known as “one of the pioneer plants” in extreme environments at glacial fronts [17]. Generally, one mode of protist succession in glacier-frontier organisms is microbial → algal lichen → moss → alpine grassland → subalpine grassland. Although soil microorganisms play a non-negligible ecological function in the process of succession, they can play a certain role in promoting the development and formation of soil as pioneer species, providing a certain material basis for the settlement, growth, and reproduction of algae and moss pioneer plants [18]. Plants and soil-microbial communities co-exist, and soil microbes interact with plants through inter-rooted or endophytic associations and promote plant growth [19]. Moss and its surrounding soil are enriched in microbial communities, which play an important ecological role in the glacial-retreat process.

Ethylene is a unique gaseous plant hormone in higher plants, and it has advantages and disadvantages for plant growth and development [20,21]. At present, various compounds have been used to control ethylene content in plants: rhizobactin and its synthetic analog aminoethoxyvinylglycine (AVG), silver thiosulfate, and 2-chloroethyl phosphoric acid. However, these chemicals are likely to cause harm to the environment, so greener methods are needed to control ethylene content in order to promote sustainable agriculture [22,23,24]. As the precursor of ethylene synthesis, 1-aminocyclopropane-1-carboxylicacid (ACC) can decompose into ammonia and α-ketobutyrate under the action of 1-aminocyclopropane-1-carboxylicacid deaminase (ACC deaminase). Several studies [25,26,27,28,29,30,31,32,33,34,35,36,37] have demonstrated that introducing ACC-deaminase-producing bacteria to a plant’s rhizosphere can decrease the amount of ethylene it produces during times of stress. This reduction in ethylene can help the plant better withstand stress. Additionally, these bacteria can utilize the α-ketobutyrate and ammonia produced by the plant to promote growth. In recent years, more and more researchers have introduced the acdS gene, i.e., the gene encoding ACC deaminase, into plant species to give them the ability to produce ACC deaminase, thereby resisting various abiotic stresses and promoting plant growth and development [38,39,40]. Therefore, ACC-deaminase-producing bacteria can be used as a green method to control ethylene produced by stress.

Therefore, we intend to use the concept of altitude instead of succession time, and use high-throughput sequencing and soil microbial screening and identification to explore how moss and its covered soil and bacterial communities in soil synergistically drive the primary succession after glacier retreat, and excavate valuable microorganisms in moss-covered soil. We also hope that through research, we can lay the foundation for the future production of biological fertilizers, the development of saline-alkali land, and the protection of glacier ecosystems.

## 2. Materials and Methods

### 2.1. Study-Area Overview and Sampling

#### 2.1.1. Study-Area Overview

Xinjiang Tianshan No. 1 Glacier is known as “the living fossil of glacier”. The average annual temperature in this region is 5.2 °C, and the negative-temperature months are 7–8 months long; January is the coldest month, with an average temperature of −15.6 °C, and July is the hottest month, with an average temperature of 4.9 °C. The average annual precipitation is 441.1 mm [18]. The climate is characterized by low partial pressure of oxygen, strong ultraviolet radiation, and strong winds. 

#### 2.1.2. Sampling Scheme

Taking the periglacial region of Glacier No. 1 of Tianshan Mountains in Urumqi, Xinjiang (3575 ± 200 m, 43°05′ N, 86°49′ E), as the sample-collection area, transects of five elevation gradients (3550–3750 m, each with a 50 m gradient) were designed. Three moss-covered soils (with aboveground plants and litter layers removed) were randomly selected, and soil samples were collected with soil drills (diameter 5 cm). The sampling depth was 0–10 cm, and three replicates were set for each plot. The soil of the parallel plots was mixed to obtain a total of 15 composite soil samples (Table 1). The composite samples were divided into three groups for the ensuing experiments: One was kept at 4 °C for the detection of physicochemical soil properties, another was sent to Beijing Novogene Co., Ltd. (Beijing, China), for high-throughput sequencing for microbial-community analysis, and the third was kept in a freezer set to 80 °C for the subsequent screening of ACC-deaminase-producing strains.

### 2.2. Determination of Physical and Chemical Properties of the Soil

Soil pH was measured using a pH meter. A soil total-potassium-content kit, soil total-phosphorus-content kit, soil available-nitrogen kit, soil available-potassium-content kit, soil organic-matter-content test kit, and neutral- and alkaline-soil available-phosphorus kits were used to measure the physical and chemical properties of the soil (this step was completed by Suzhou Comin Biotechnology Co., Ltd., Suzhou, China).

### 2.3. High-Throughput Sequencing of the Soil Samples

To perform high-throughput sequencing, BeijingNovogene Co., Ltd., received the pretreatment soil-sample packets. The lower machine’s raw data were stitched together and cleaned up for further analysis. Based on the clean data, clustering, and species classification, analyses of OTUs were carried out. The results of the OTU clustering show that the representative sequences for each OTU underwent species annotation. OTUs were also evaluated for Venn diagrams, alpha-diversity calculations, and abundance. In addition, a correlation analysis of environmental factors was performed.

### 2.4. Isolation and Purification of ACC-Deaminase-Producing Strains

A 1 g soil sample was weighed and placed in 100 mL PAF medium and cultured at 30 °C 180 r/min for 24 h; then, 1 mL was added to 50 mL DF liquid medium and cultured under the same conditions for 24 h. Then, 1 mL of the above culture medium was taken, added to 50 mL ADF liquid medium, and cultured at 30 °C 180 r/min for 48 h. After gradient dilution, 200 μL were taken for each gradient, spread on an ADF plate, and cultured at 30 °C for 48~72 h, and the results were observed; then, the strains with different colony morphologies were selected for numbering and streak purification, and the colony morphologies of the purified strains were recorded [41].

The purified strain was cultured with TSB culture medium at 30 °C 180 r/min until turbid, and 1 mL of bacterial solution was added to 1 mL of 20% sterile glycerol and mixed well in a cryotube and stored at −80 °C.

### 2.5. Determination of ACC-Deaminase Activity 

ACC was hydrolyzed by ACC deaminase into α-ketobutyrate and ammonia. According to Penrose et al. [41], α-ketobutyrate was stained using the 2,4-dinitrophenylhydrazine reagent to determine the content of α-ketobutyrate in the bacterial solution, thereby illustrating the ACC-deaminase activity of the strain. Bovine serum albumin was used as a standard protein, and protein content was detected using the Coomassie Blue (Bradford) assay. Each assay was repeated three times. The ACC-deaminase activity was expressed as the amount of α-ketobutyrate produced per mg of protein per minute.

### 2.6. Strain Identification

The isolated strains with high activity were screened for colony-morphology observation, and the physiological and biochemical tests and methods were based on the physiological- and biochemical-determination parts related to bacteria in the Bergey’s Manual of Determinative Bacteriology, 7th ed. [42].

### 2.7. BIOLOG Analysis

Potential carbon-source utilization and the degree of sensitivity to chemicals of the isolates were assessed using BIOLOG GEN III MicroPlates according to the manufacturer’s instructions. The reactions were observed after 24 h incubation at 30 °C and read using an automated Biolog MicroStation Reader.

### 2.8. Sequencing of 16S rDNA Gene for Identification of Isolated Strains

After isolation and purification, the strains were stored in TSB liquid and sent to Shanghai Sangon Biotech Company at −20 °C to complete the determination of the 16S rDNA sequence. The 16S rDNA gene sequences of the related strains with the highest similarity were found in the GenBank database for comparative BLAST analysis. Phylogenetic trees were created using the neighbor-joining method in MEGA11 software (version 11.0.11).

### 2.9. Statistical Analyses

All samples were analyzed in triplicate, and the data were presented as the mean ± the standard deviation for each sample point. All data were collected to analyze the variance at *p* < 0.05, and Duncan’s multiple-range test was applied to compare the mean values. Data-processing and -graphing software is available for GraphPad Prism (version 8.2.1 (441)), IBM SPSS Statistics 20 (version 20), Molecular Evolutionary Genetics Analysis (version 11.0.11).

## 3. Results

### 3.1. Physical and Chemical Properties of Moss-Covered Soil at Different Altitudes

The physical and chemical properties of moss-covered soils at different altitudes are shown in Table 2. As can be seen from the table, there was no statistically significant difference in pH, TN, AN, or TP content, and there was a statistically significant difference in TK, AP, AK, and SOM content in moss-covered soils at different altitudes, mainly between the AY3550 transect and other transects. There was a significant difference of a 32.27% decrease in soil TK from the BY3750 transect to the AY3550 transect. There was a significant difference of a 33.36% decrease in soil AP from the BY3750 transect to the AY3550 transect. There was a significant difference of a 56.61% decrease in soil AK from the BY3750 transect to the AY3550 transect. There was a significant difference of an 83.48% decrease in soil AP from the BY3750 transect to the AY3550 transect.

### 3.2. Data Pre-Processing Statistics, Quality Control, and Distribution of OTUs in Moss-Covered Soils at Different Altitudes

Based on the high-throughput-sequencing results (Table 3), the interval of clean reads of 15 soil samples ranged from 52,994 to 68,817, the interval of effective sequences accounted for the total sequences from 75.29% to 79%, and the range of the total number of sample OTUs was from 3230 to 4476. From the species-accumulation boxplot (Figure 1), it can be seen that the location of the box plot tended to be flat, indicating that species in this environment did not increase significantly with increasing sample size, indicating adequate sampling and data analysis.

The number of common and unique bacterial OTUs in moss-covered soils at five different altitudes was analyzed using a Venn diagram, and the overlap of bacterial OTUs was visually displayed. In five transects with different altitudinal gradients, the number of common bacterial OTUs of moss-covered soils was 2852, and the numbers of unique bacterial OTUs were as follows: AY3550, 1278; EY3600, 664; DY3650, 586; CY3700, 641; BY3750, 479. From Figure 1b, it can be seen that there were differences in the distribution of bacterial OTUs in moss-covered soil at different altitudes, and the main differences were reflected in the AY3550 transect and other transects. We found a 1.67-fold increase in soil-bacteria-specific OTUs from the highest to the lowest altitude. Based on the OUT distribution of the Venn graph, five transects could be divided into three groups: an AY3550 group, a BY3750 group, and a group with the three other transects. In our results (Figure 1, Table 3) we found that the total number of OTUs in each band tended to increase with decreasing elevation, and among them the number of band-specific OTUs tended to increase, then decreased and increased.

### 3.3. Bacterial Diversity and Community Composition

By analyzing the relative abundance of bacterial phyla in moss-covered soil at different altitudes, it can be seen that each altitude had similar phyla but the relative abundance was different. A total of 10 dominant phyla with relative abundances ≥ 1% were detected in moss-covered soils (Figure 2a). Among the top 10 dominant phyla in relative abundance at the bacterial-phylum level in moss-covered soils at different altitudes, Proteobacteria (21.40–27.26%), Acidobacteriota (10.41–14.63%), and Actinobacteriota (9.05–11.76%) had the highest relative abundance. Proteobacteria and Myxococcota showed an increasing–decreasing–increasing trend in richness with decreasing altitudinal gradient. Acidobacteriota and Nitrospirota showed an increasing–decreasing trend in richness with decreasing altitudinal gradient. Verrucomicrobiota, Chloroflexi, and Firmicutes showed a decreasing–increasing–decreasing trend in richness as the altitudinal gradient decreased. Actinobacteriota showed a decreasing–increasing trend in richness with decreasing altitudinal gradient. In addition, Bacteroidota showed an increasing trend in richness with decreasing altitudinal gradient.

From the heat-map analysis of dominant bacterial communities at the level of the first 35 genera of moss-covered soil, it can be seen that the composition of bacterial communities in moss-covered soil changed significantly with different altitudes. In the bacterial community in moss-covered soil, 17 species clustered abundantly in soil samples from the AY3550 group, 10 species clustered abundantly in soil samples from the EY3600 group, 9 species clustered abundantly in soil samples from the DY3650 group, 12 species clustered abundantly in soil samples from the CY3700 group, and 7 species clustered abundantly in soil samples from the BY3750 group (Figure 2b). Cluster analysis showed that the trend of variation in the bacterial-community structure and quantity in moss-covered soil was different at different altitudes, and there were significant differences in molecular polymorphisms. In moss-covered areas, the structure and number of bacteria at the altitude of AY3550 were the most different from those at other altitudes, and DY3650 and BY3750 tended to be similar and grouped together, although there was a gap with samples of EY3600 and CY3700, in which the difference was not as obvious as in AY3550.

### 3.4. Analysis of the Diversity Index of Moss-Covered Soils at Different Altitudes

An analysis of the diversity of soil-bacterial communities covered with moss at various altitudes was carried out, and the results of the number of OTUs, Shannon index, Simpson index, ACE index, and Chao1 index of soil-bacterial communities at different altitudes are shown in Table 4. From multiple diversity indices, the Shannon index and Simpson index of bacteria were not significantly different in moss-covered soil at different altitudes. However, the Shannon and Simpson indices of bacteria increased with decreasing altitude. In addition, there were significant differences in the Chao1 and ACE indices of bacteria, mainly in the Chao1 index of moss-covered soil, which was significantly different between the AY3550 transect and the BY3750 transect, and the ACE index of moss-covered soil was significantly different between the AY3550 transect, the BY3750 transect, and the EY3600 transect (*p* < 0.05). Interestingly, the moss-covered soil-bacterial Chao1 index and ACE index showed a trend of increasing, then decreasing, and then increasing with decreasing altitude.

### 3.5. Analysis of Bacterial-Community Structure and Environmental Factors in Moss-Covered Soils at Different Altitudes

The results of principal coordinate analysis of soil-bacterial communities covered with moss at different altitudes are shown in Figure 3a, and the more similar the community composition of the samples the closer they are to each other in the figure. In this paper we performed PCoA analysis based on weighted Unifrac distance. PC1 and PC2 explained 40.97% and 15.78% of the sample differences, respectively, totaling 56.75%. From the figure, it can be seen that the sample strips EY3600, DY3650, CY3700, BY3750, and AY3550 had obvious distribution differences of moss-covered soil. Based on the results of soil physicochemical properties at different elevations, we selected four significantly different environmental factors, TK, AK, SOM, and AP, for RDA analysis, which showed that environmental factors explained 38.11% on the first axis of the RDA biplot and 22.72% on the second axis (Figure 3b). Microbial communities were mainly influenced by SOM (R^2^ = 0 0.7752), TK (R^2^ = 0.7651), AP (R^2^ = 0.6609), and AK (R^2^ = 0.7364) (Figure 3b).

Spearman-correlation analysis can be used to identify correlations between environmental factors and groups. The species with the top 35 relative abundances were selected for Spearman-correlation analysis with environmental factors. At the genus level (Figure 4), the species of soil-bacterial community covered by Tianshan No. 1 glacial moss had different degrees of correlation with environmental factors, of which altitude showed a significant positive correlation with *Candidatus_Udaeobacter* and a significant negative correlation with *Oryzihumus*, *Ellin6067*, *MND1*, and *Ramlibacter* (*p* < 0.01); AN, SOM, and TN showed a significant positive correlation with *Solrobacter* and *Pseudonocardia* (*p* < 0.01); and SOM showed a significant negative correlation with Subgroup 10 (*p* < 0.01). At the genus level, elevation and SOM may be more relevant to moss-covered soil-bacterial communities than other environmental factors. Additionally, there were positive correlations between *Pseudonocardia* and various environmental factors (TK, AP, TN, AK, AN, TP, SOM, and altitude).

### 3.6. Isolation and Purification of ACC-Deaminase-Producing Strains from Moss-Covered Soil and Determination of ACC-Deaminase Activity

A total of 33 strains of ACC-deaminase-producing bacteria were isolated and purified from 15 soil samples at different altitudes, including 7 strains of AY3550, 7 strains of EY3600, 7 strains of DY3650, 7 strains of CY3700, and 5 strains of BY3750. The regression equation for the α-ketobutyrate standard curve was Y = 2.649X + 0.0237 (R^2^ = 0.9988), and the regression equation for the bovine-serum-albumin standard curve was Y = 0.8212X + 0.019 (R^2^ = 0.9976).

Enzyme-activity-assay results for 33 ACC-deaminase-producing strains after three repeated measurements are shown in Figure 5. We found that there were differences in enzyme activity among 33 strains of bacteria at different altitudes, and the difference was statistically significant (*p* < 0.05). The overall trend of enzyme activity showed a tendency to increase and then decrease with decreasing altitude. Through data processing and statistical analysis, we found that the enzyme activities of DY1–3, DY1–4, and EY2–5 strains were the highest, at 4.733 5 ± 0.036 5 U/mg, 4.579 5 ± 0.109 5 U/mg, and 4.565 5 ± 0.044 5 U/mg, respectively, and we subsequently selected the three strains with the highest enzyme activities for the study.

### 3.7. Identification of Three Strains with High ACC-Deaminase Activity

#### 3.7.1. Colony Morphology and Microscopic Observation

As can be seen from Figure 6, DY1–3 and DY1–4 colonies showed yellowish, opaque, moist round bulges, which were Gram-negative rods, and EY2–5 colonies showed yellowish opaque moist oval bulges, which were Gram-negative rods.

#### 3.7.2. Physiological- and Biochemical-Test Results

Table 5 indicates that all three strains exhibited acid production, utilized malonic acid and citric acid, hydrolyzed starch, tested positive for methyl red and urease, tested negative for V-P, and produced indoleacetic acid.

#### 3.7.3. Molecular-Biology Identification of PGPR with ACC-Deaminase Activity

The research (Figure 7) shows that the best-matching names of DY1–3, DY1–4, and EY2–5 in the ID address bar were as follows: DY1–3, *Pseudomonas fluorescens* (PROB = 0.555, SIM = 0.555 > 0.5); DY1–4, *Pseudomonas fluorescens* (PROB = 0.596, SIM = 0.596 > 0.5); and EY2–5. *Pseudomonas fluorescens* (PROB = 0.589, SIM = 0.589 > 0.5). The three strains showed different utilization of 71 different carbon sources on GEN III microplates, and the corresponding metabolic fingerprints also showed some differences. As seen in Figure 7a, strain DY1–3 detected 24 nitrogen sources, such as α-D-glucose, β-hydroxy-phenylacetic acid, γ-amino-butyric acid, D-fructose, and gelatin, available after 24 h incubation; it was chemically sensitive to 13 substances, such as 1% sodium lactate, troleandomycin, and lincomycin. After 24 h incubation of strain DY1–4, 26 nitrogen sources, such as α-D-glucose, β-hydroxy-phenylacetic acid, D-galacturonic acid, D-mannitol, and D-trehalose were detected to be available; 13 substances, such as 1% sodium lactate, troleandomycin, and lincomycin were chemically sensitive. After 24 h incubation of strain EY2–5, 23 nitrogen sources such as α-D-glucose, D-fructose, D-mannitol, D-mannose, and L-arginine were detected to be available; 10 substances, such as 1% sodium lactate, troleandomycin, and lincomycin, were chemically sensitive.

According to 16S rDNA-sequence determination, the DNA base pairs of DY1–3, DY1–4, and EY2–5 strains were 1357 bp, 1351 bp, and 1357 bp, respectively. The nucleic-acid-sequence (BlastN) results of DY1–3, DY1–4, and EY2–5 strains were highly similar to those of *Pseudomonas* spp. (Table 6). The phylogenetic tree (Figure 8) showed that DY1–3 and DY1–4 clustered in the same clade with sequence number KR085873 and sequence number KR085942, with 82% similarity, indicating that DY1–3 and DY1–4 were the same bacteria and a subspecies of *Pseudomonas cedrina* subsp. *Cedrina*. The EY2–5 strain was similar to 79% of strain JX827233 and was identified as *Pseudomonas lurida*. Combined with the above results, it was identified that all three strains may be *Pseudomonas* spp.

## 4. Discussion

Our study found that the content of SOM varied most significantly in different elevations, and the SOD content of the AY3550 sample zone was significantly different from other sample zones (*p* < 0.05); meanwhile, the SOM content showed a decreasing trend with succession, which may have been due to the increase in soil-microbial biomass with succession and the decrease in the binding capacity of soil particles to SOM, which made SOM lack physical protection, and it was decomposed into DOM by microorganisms [43]. However, dissolved organic matter (DOM), as the most active organic component in soil, plays a critical role in soil biogeochemical processes. The content of P and K in soil is higher at the early stage of soil development and decreases with the increase in succession time, which may be mainly due to the release of mineral-nutrient elements due to the weathering of minerals at the early stage of soil development, whereas it is affected by vegetation and exogenous input at the later stage [44]. There was no significant difference in pH, TN, AN, or TP content in moss-covered soils, but there were still differences, which were inconsistent with previous studies and may be because mosses accelerated soil carbon, nitrogen, and phosphorus cycling and significantly improved soil properties in their vital activities [45,46].

Soil microorganisms are extremely important and active parts of soil ecosystems and dominate soil-nutrient transformation cycles, system stability and resistance to disturbances, and sustainable soil productivity [47]. Our results showed that the number of unique OTUs of moss-covered soil bacteria decreased with increasing elevation, and five transects could be divided into two successional stages according to the number of unique OTUs: the AY3550 transect and other transects. With the succession stage, there was no significant difference in the Shannon index or Simpson index, but the ACE index and Chao1 index increased significantly (*p* < 0.05). The community-diversity index showed a trend of first increasing, then decreasing, and then increasing with altitude, and the overall performance was 3550 m > 3700 m > 3650 m > 3600 m > 3750 m. Possible reasons for such a situation In general are that as the altitude rises, the succession time becomes shorter, and the surrounding environment becomes relatively poor, and as the environment becomes more extreme, the overall microbial diversity diminishes and a few groups dominate [48,49,50]. At the phylum level, we found that Proteobacteria and Acidobacteria consistently dominated during successional stages, and the relative abundance of Verrucomicrobia decreased, whereas that of Actinobacteria increased. With succession time, there were some differences in relative abundances at the phylum level: Proteobacteria and Actinobacteriota showed an increasing trend as a whole, and Acidobacteriota and Verrucomicrobiota showed a decreasing trend as a whole. Additionally, we found that the dominant phyla were Proteobacteria, Acidobacteriot, and Verrucomicrobiota at the early stage of moss-covered soil development, whereas the dominant phyla were Proteobacteria, Acidobacteriota, and Actinobacteriota at the late stage of development, and the richness of Verrucomicrobiota decreased significantly. The relative abundance of Actinobacteria increased, mainly in association with the increase in animal and plant remains in successional soils. The occurrence of differences in the abundance of dominant bacteria at different successional stages may be mainly influenced by changes in soil-nutrient content [51,52].

Principal coordinate analysis (PCoA) is used to extract the most dominant elements and structures from multidimensional data by a series of feature values and feature-vector ordering. In this paper, we performed PCoA based on the weighted Unifrac distance. If the samples in the PCoA diagram are closer, it indicates that the sample-community composition is more similar. In this study, we found that PCoA divided the AY3550 transect and the other four transects into two parts, with large differences in community composition, whereas EY3600, DY3650, CY3700, and BY3750 clustered together in the figure, with small differences in community composition. The clustering heat map at the genus level likewise divided the AY3550 transect into two branches from the other four transects. According to the results of PCoA analysis and the clustering heat map, the succession of soil bacteria covered with moss in Tianshan No. 1 Glacier could be divided into two stages: early stage (EY3600, DY3650, CY3700, BY3750) and late stage (AY3550). Similar results were reported by Wu et al. [53] and Huang et al. [54]. The results of RDA and Spearman analysis showed that SOM was the main environmental factor affecting the diversity and structure of soil communities covered by moss in Tianshan No. 1 Glacier. Soil organic matter plays an important role in the formation of soil aggregates. We suggest that as glaciers recede, rock plants colonize and grow in periglacial areas, leading to succession and affecting the physical and chemical properties of soils and microfaunal communities. These changes in soil properties affect the abundance and community structure of microbial communities, ultimately contributing to plant-community succession throughout the periglacial region.

As one of the pioneer plants in the extreme environment of the glacier front, moss plants, in addition to their own functions of soil formation, water retention, and nitrogen fixation, also play an important role in the rich microbial community on their body surface and in the surrounding soil, and synergistically drive the plant-community succession in the ice-margin area. ACC-deaminase-producing PGPR has been considered in recent years an effective plant probiotic in plant roots. Several studies [22,23,24,38,39,40,55,56,57,58,59] have shown that bacteria that produce ACC deaminase and are introduced to the rhizosphere of plants can decrease the amount of ethylene produced by plants under stress. This reduction helps plants resist stress while also promoting growth by using the produced α-ketobutyrate and ammonia. In this experiment, 33 strains of ACC-deaminase-producing bacteria were isolated from moss-covered soil samples collected from different altitudes. The enzyme activity ranged from 0.067 U/mg to 4.7375 U/mg. Some scholars pointed out that when the ACC deaminase activity was higher than 0.020 U/mg, it promoted plant growth [60]. In this experiment, the 33 preliminarily isolated strains facilitated plant growth, and the enzyme activities of 13 strains exceeded 2 U/mg, with the enzyme activity of the DY1–3, DY1–4, and EY2–5 strains exceeding 4.5 U/mg. In some scholars’ studies [27,34,61,62], the activity of ACC-deaminase-producing strains isolated from this experiment was also determined, and there were differences in the size of the enzyme activity. Through comparison, it can be concluded that the ACC-deaminase-producing activity of the strains obtained in this experiment was high and had high research value. In the process of bacterial identification, this experiment combined traditional identification methods based on the Biolog detection method using carbon and nitrogen sources with the currently widely used 16S rDNA identification method. Biolog identification results identified three strains of DY1–3, DY1–4, and EY2–5 as *Pseudomonas fluorescens*. However, 16S rDNA-sequencing results identified DY1–3 and DY1–4 as *Pseudomonas cedrina subsp. cedrina* and EY2–5 as *Pseudomonas lurida*. The three methods mutually confirmed the identification of DY1–3, DY1–4, and EY2–5 as *Pseudomonas*. The plant-growth-promoting bacteria producing ACC deaminase isolated [26,27,29,31,32,63,64] at home and abroad are mainly *Pseudomonas* and *Bacillus*, which are inseparable from *Pseudomonas* as indigenous microorganisms in nature (especially in soil), and their ranges of changes in growth temperature and pH value are large. Furthermore, the types of nutrient metabolism are diverse, and *Pseudomonas* producing ACC deaminase is important in beneficial plant–microorganism interaction [26]. At present, *Pseudomonas* plays an increasingly important role in environmental bioremediation, biological control, biotransformation, and other fields. In this experiment, the three strains were found to have many plant-growth-promoting characteristics, such as IAA production and ACC deaminase production, which may have significant growth-promoting and stress-resistance-enhancing effects on plants, excellent material decomposition and utilization ability, and high research value.

The enzyme activities of ACC-deaminase-producing bacteria screened from soil samples covered in moss at various altitudes were noticeably different [27,34,61,62], which was strongly associated with the succession of periglacial zones following glacier degeneration. We concluded that the BY3750 transect is undergoing the early stages of enamel degradation and melting and that the environment is unfavorable. Because it has just recently started to colonize, the moss has not yet developed an adaptation to its surroundings. There are no circumstances that would allow the matching PGPR to survive, and the DY3600 transect normally has strong enzyme activity. We think that as succession changes, bryophytes establish themselves and expand in the periglacial region using their own mechanisms for soil formation, water retention, nitrogen fixation, and other processes.

## 5. Conclusions

In conclusion, under the influence of global warming, glaciers have retreated significantly, resulting in the exposure of glacial soils and the subsequent onset of succession. In this study, we suggest that mosses, soils, and plant-promoting bacteria act synergistically to drive succession during the process. As the succession continues, the soil’s physicochemical properties vary under the influence of moss plants’ colonization, growth, and development, which lead to changes in the bacterial-community abundance in the soil microcosm, resulting in the enrichment of ACC-deaminase-producing phytopathogenic bacteria with different enzymatic activities under moss-covered soil at different stages of succession.

## Figures and Tables

**Figure 1 microorganisms-11-01521-f001:**
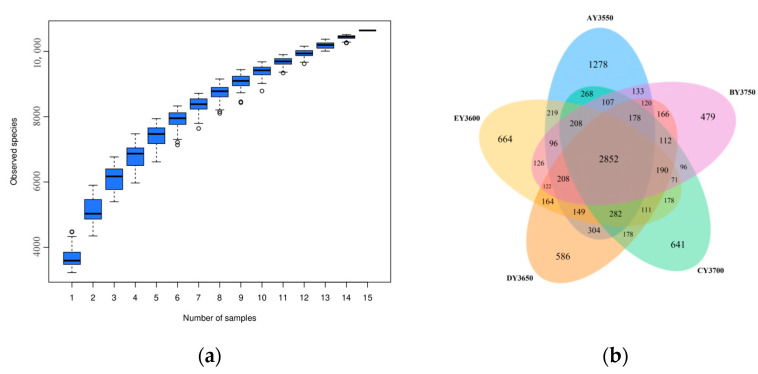
(**a**) Species-accumulation boxplot of moss-covered soils at different altitudes; (**b**) distribution of OTUs in moss-covered soils at different altitudes.

**Figure 2 microorganisms-11-01521-f002:**
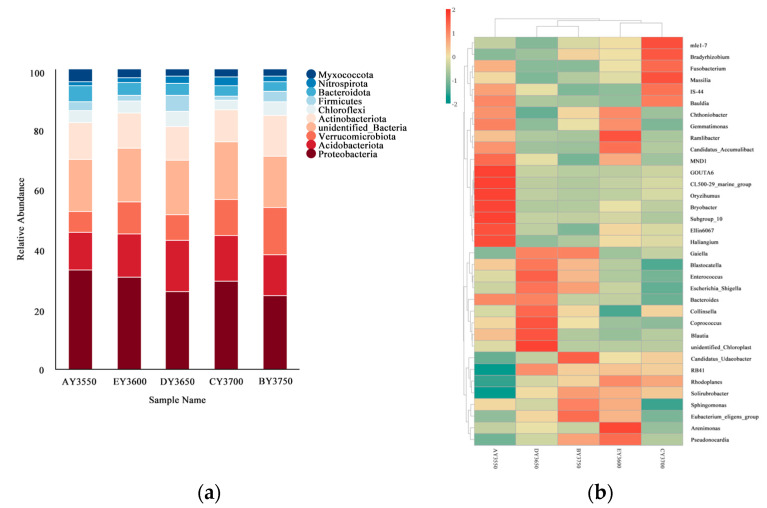
(**a**) Relative abundance of the bacterial community at the phylum level in moss-covered soils at different altitudes; (**b**) clustering heat map of the bacterial community at the genus level in moss-covered soils at different altitudes.

**Figure 3 microorganisms-11-01521-f003:**
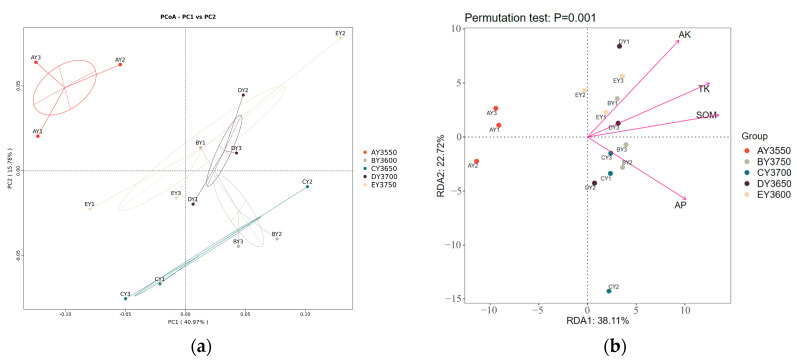
(**a**) PCoA of moss-covered soil bacterial communities at different altitudes; (**b**) RDA of moss-covered soil-bacterial communities at different altitudes.

**Figure 4 microorganisms-11-01521-f004:**
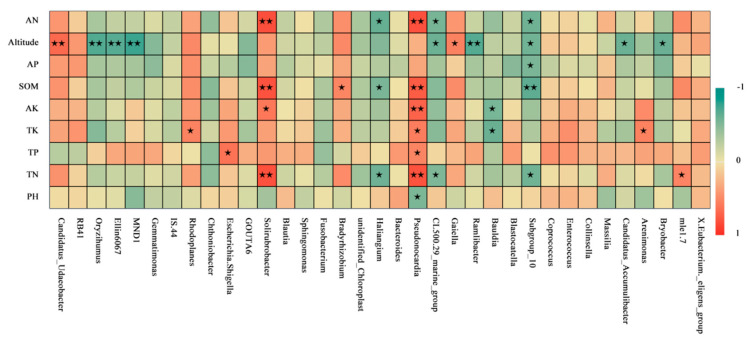
Spearman-correlation analysis heat map based on the bacterial and environmental factors in moss-covered soils at different altitudes. (★ represents when *p* < 0.05 and bacterial-community species are correlated with environmental factors; ★★ represents when *p* < 0.01 and bacterial-community species are correlated with environmental factors).

**Figure 5 microorganisms-11-01521-f005:**
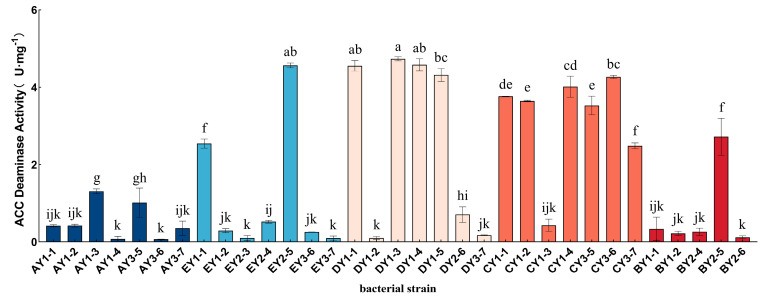
Determination of ACC-deaminase activity of 33 isolated and purified strains. (The lowercase letters in the graph indicate significant differences in enzyme-activity size among the 33 strains obtained from screening in moss-covered soil (*p* < 0.05).)

**Figure 6 microorganisms-11-01521-f006:**
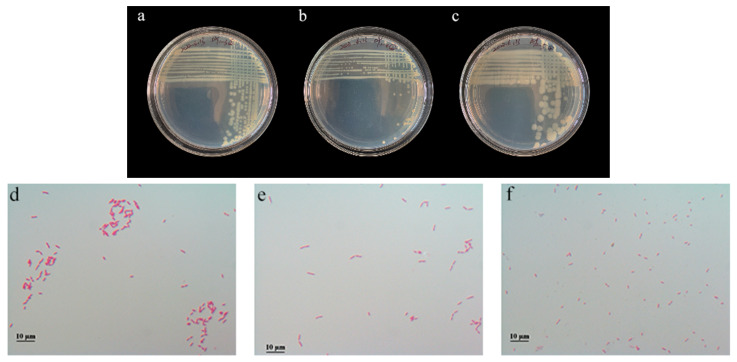
Colony morphology and Gram-stain results for DY1–3, DY1–4, and EY2–5. (**a**,**d**) DY1–3; (**b**,**e**) DY1–4; (**c**,**f**) EY2–5.

**Figure 7 microorganisms-11-01521-f007:**
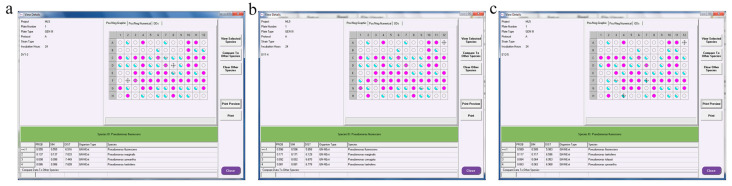
Biolog identification results for DY1–3, DY1–4, and EY2–5 strains. (**a**) DY1–3; (**b**) DY1–4; (**c**) EY2–5).

**Figure 8 microorganisms-11-01521-f008:**
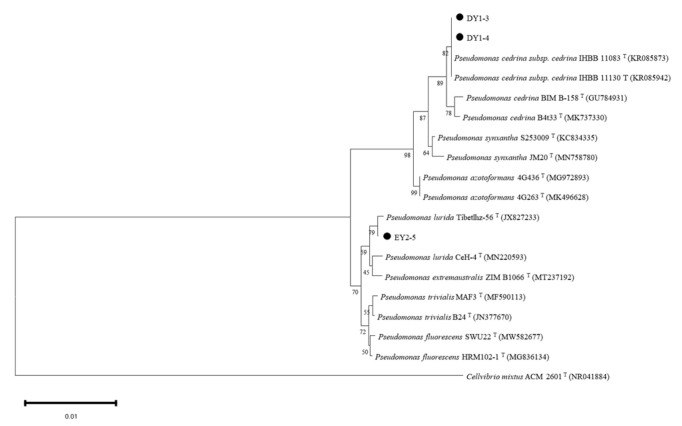
Sequence-phylogenetic tree of 16S rDNA of three strains.

**Table 1 microorganisms-11-01521-t001:** Detailed soil samples were collected in the periglacial region of Glacier No. 1 in the Tianshan Mountains.

Soil Transect	Soil Sample No.	Altitude	Latitude and Longitude	
AY3550	AY1, AY2, AY3	3550 ± 10 m	86°49′42″ E	43°06′36″ N
EY3600	EY1, EY2, EY3	3600 ± 10 m	86°49′43″ E	43°06′40″ N
DY3650	DY1, DY2, DY3	3650 ± 08 m	86°49′31″ E	43°06′05″ N
CY3700	CY1, CY2, CY3	3700 ± 10 m	86°49′31″ E	43°06′55″ N
BY3750	BY1, BY2, BY3	3750 ± 04 m	86°49′24″ E	43°07′01″ N

**Table 2 microorganisms-11-01521-t002:** Physical and chemical properties of moss-covered soil at different altitudes.

Belt Transect	pH ^1^	TK ^2^ (g/kg)	AP ^3^ (mg/kg)	TN ^4^ (g/kg)	AK ^5^ (mg/kg)	AN ^6^ (mg/kg)	TP ^7^ (g/kg)	SOM ^8^ (%)
AY 3550	6.83 a	11.42 b	8.05 b	0.8 a	85.84 b	36.17 a	0.5653 a	1.88 b
EY 3600	6.80 a	18.03 a	12.79 ab	4.73 a	320.97 a	158.08 a	0.5780 a	9.62 a
DY 3650	6.77 a	18.34 a	11.30 ab	4.39 a	302.92 ab	171.50 a	0.5960 a	10.03 a
CY 3700	6.82 a	15.47 ab	15.26 a	6.71 a	187.44 ab	166.83 a	0.4824 a	10.84 a
BY 3750	6.85 a	16.86 a	12.08 ab	5.36 a	197.85 ab	175.00 a	0.5568 a	11.38 a

^1^ pH: soil pH; ^2^ TK: soil total-potassium content; ^3^ AP: soil available-phosphorus content; ^4^ TN: soil total-nitrogen content; ^5^ AK: soil available-potassium content; ^6^ AN: soil available-nitrogen content; ^7^ TP: soil total-phosphorus content; ^8^ SOM: soil organic-matter content; different lowercase letters in the same column indicate significant differences between moss-covered soils at different altitudes (*p* < 0.05).

**Table 3 microorganisms-11-01521-t003:** Summary of statistics and quality control for data preprocessing.

Sample Name	Raw Reads ^1^	Clean Reads ^2^	GC% ^3^	Effective% ^4^	OTUs
AY1	82,141	63,534	56.05	77.35	4331
AY2	79,582	62,640	56.05	78.71	4476
AY3	86,249	65,066	55.98	75.44	3927
EY1	74,159	58,582	56.21	79.00	3567
EY2	78,425	60,570	56.65	77.23	3477
EY3	81,972	64,192	56.28	78.31	3779
DY1	85,981	65,597	56.38	76.29	3777
DY2	74,371	56,969	56.75	76.60	3644
DY3	80,684	63,269	56.75	78.42	3517
CY1	69,956	52,994	56.64	75.75	3852
CY2	74,920	58,855	56.34	78.56	3476
CY3	87,870	67,659	56.44	77.00	3582
BY1	80,949	62,234	56.59	76.88	3595
BY2	83,779	65,904	56.71	78.66	3230
BY3	91,402	68,817	56.31	75.29	3275

^1^ Raw reads: filters out the sequence of low-quality bases; ^2^ clean reads: sequence for subsequent analysis; ^3^ GC (%): the number of bases in clean reads; ^4^ effective (%): percentage of number of clean reads to number of raw reads.

**Table 4 microorganisms-11-01521-t004:** Index of bacterial-community diversity in moss-covered soils at different altitudes.

Sample	OTUs *	Shannon *	Simpson *	Chao1 *	ACE *	Goods Coverage *
AY3550	4244 a	10.1670 ± 0.2440 a	0.9967 ± 0.0009 a	4715.777 ± 298.897 a	4811.849 ± 315.213 a	0.981
EY3600	3607 b	9.8241 ± 0.1800 a	0.9958 ± 0.0014 a	3975.773 ± 164.250 ab	4043.742 ± 197.426 b	0.985
DY3650	3646 b	9.8191 ± 0.1035 a	0.9959 ± 0.0010 a	4034.307 ± 204.338 ab	4117.463 ± 222.323 ab	0.985
CY3700	3636 b	9.5642 ± 0.2250 a	0.9933 ± 0.0021 a	4159.167 ± 426.325 ab	4268.216 ± 409.202 ab	0.982
BY3750	3366 b	9.5212 ± 0.3929 a	0.9925 ± 0.0056 a	3719.790 ± 224.612 b	3814.941 ± 197.318 b	0.986

* Different lowercase letters in the same column indicate significant differences between moss-covered soils at different altitudes (*p* < 0.05).

**Table 5 microorganisms-11-01521-t005:** Physiological and biochemical results of three ACC-deaminase-producing strains.

Test Items	Strains *		
	DY1–3	DY1–4	EY2–5
Litmus milk test	AO	AO	AO
Malanate-utilization test	+	+	+
Citrate-salt test	+	+	+
Hydrolysis of starch	+	+	+
Methyl-red test	+	+	+
Urease test	+	+	+
Indole production	+	+	+
V-P test	-	-	-
Litmus milk test	AO	AO	AO
Malanate-utilization test	+	+	+

* AO: acid output; “+”: positive; “-”: negative.

**Table 6 microorganisms-11-01521-t006:** Nucleic-acid-sequence (BlastN) alignment results for DY1–3, DY1–4, and EY2–5 strains.

Strain	Description of the Strain with the Highest Homology (Login Number)	Max Score	Total Score	Query Cover	E Value	Ident
DY1–3	*Pseudomonas cedrina* subsp. *cedrina* strain IHBB 11130 16S ribosomal RNA gene, partial sequence (KR085942.1)	2507	2507	100%	0.0	100%
DY1–4	*Pseudomonas cedrina* subsp. *cedrina* strain IHBB 11130 16S ribosomal RNA gene, partial sequence (KR085942.1)	2507	2507	100%	0.0	99.78%
EY2–5	*Pseudomonas* sp. strain 33R 16S ribosomal RNA gene, partial sequence (MK070918.1)	2505	2505	100%	0.0	100%

## Data Availability

The data presented in this study are available on request from the corresponding author.

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
