# Peer review of "Bacterial Diversity Analysis and Screening for ACC Deaminase-Producing Strains in Moss-Covered Soil at Different Altitudes in Tianshan Mountains—A Case Study of Glacier No. 1"

_microorganisms, 2023, doi:10.3390/microorganisms11061521_

Round 1
Reviewer 1 Report
Scientific article with basic research in the field of environmental microbiology.
The introduction introduces the topic well. The purpose of the work is well defined. The chapter materials and methods is very detailed, the composition of microbiological media seems unnecessary. The results of the research are presented very well, the graphics are accurate, careful and legible. The discussion and short conclusions result from the presented material.
Author Response
Dear reviewer:
Thank you for your comments concerning our manuscript entitlerd "Bacterial diversity analysis and screening for ACC deaminase-producing strains in moss-covered soil of Glacier No. 1 at different altitudes in Tianshan Mountains" (ID: 2415541). First of all, thank you very much for your approval of this article and for your valuable comments. This comment is valuable and very helpful for revising and improving our paper, as well as the important guiding significance to our researches. We have studied this comment carefully and have made correction which we hope meet with your approval. The main corrections in the paper and the responds to the comments are as flowing.
1.Response to comment: The chapter materials and methods is very detailed, the composition of microbiological media seems unnecessary.
Response: Deleted microbiological media.
Special thanks to you for your good comments.
Sincerely,
Corresponding author:
Name: Yonghong Fan E-mail: fyh@xju.edu.cn

Reviewer 2 Report
Dear authors, the manuscript entitled "Bacterial diversity analysis and screening for ACC deaminase-producing strains in moss-covered soil of Glacier No. 1 at different altitudes in Tianshan Mountains" presents an interesting approach on a topic close related to current climate changes and the effect on bacterial diversity in one extreme environment. There are some suggestions that I consider will improve the current form of the manuscript.
Title - you can change galcier no. 1 to something more general like "a glacier", or you can change the title to: Bacterial diversity analysis and screening for ACC deaminase-producing strains in moss-covered soil at different altitudes in Tianshan Mountains - a case study of Glacier No. 1"
All the text - pay attention to Latin names of genus and species to be written in italic (e.g. line 49)
Keywords - change the keywords that are present in the Title with other similar ones. In this way, you can expand the list of words/concepts related to your work.
Introduction - in the first paragraph, more international references to sustain the problem of climate changes and their effect on glaciers. Also, you can add one or two sentences to describe the glaciers as unique ecosystems. This is vital for the importance of such type of studies.
Try to use grassland instead of meadow in the text
The last paragraph of the introduction should present a clear aim of the research, with separate sentences for objectives/hypotheses proposed by the authors.
Material and methods
It would be interesting to make a sampling scheme where to point visually the sampling procedure. This will help other researchers to replicate this type of research.
Add the reference for Manual of Systematic Identification of Common Bacteria and Microbiological Experiments - lines 156-157
Add a data analysis subsection to explain which type of test are used, and why, to analyze the results. Also, the software is important to be presented and referenced.
Results section
Expand the interpretation of Table 2 and use the words significant (where is the case) to analyze the differences between sampled location.
Expand the interpretation of Table 3 and Figure 1. You have interesting results, they need to be fully presented.
Table 4 - similar suggestion as above.
Figure 3 -the variance explained by PCA and RDA is low. Expand the interpretation and analyze if the low variance o not affect the quality of results presentation. It can be used other ordination as NMDS or PCoA if they present clearer the results. Or, try to make another PCA and RDA, additional to the ones already present where to exclude AY samples. Figure 3 will have 4 graphs, and you can discuss on the differences between communities and within communities. I suggest the last solution. This will add more results to be presented.
Subsection 3.6. Figure 5 do not give much information to increase the quality of the study. It can be removed.
Expand the interpretation of Figure 6. Here you need to point with more explanations why did you choose the three strains to further test.
Expand the interpretation of Biolog identification results (3.7.3.). Analyze the different use of substrates and the specific patterns for each. This will help you to complete the Discussion section. Connect the 3.7.3 and 3.7.4. subsections.
Discussion section
First paragraph is more useful in the Introduction (Lines 346-351) it does not help in this section.
Overall, this section link well the results with international references.
Rewrite the sentences from 456-463 to exclude the word such speculation. Try to identify references that sustain this affirmation or to propose a new explanation. Use words such "a possible explanation", "this phenomenon was caused by" or something similar.
Overall, the manuscript is interesting and brings new ideas, which can be further developed in similar studies.
Author Response
Dear reviewer:
Thank you for your comments concerning our manuscript entitlerd "Bacterial diversity analysis and screening for ACC deaminase-producing strains in moss-covered soil of Glacier No. 1 at different altitudes in Tianshan Mountains" (ID: 2415541). Those comments are all valuable and very helpful for revising and improving our paper, as well as the important guiding significance to our researches. We have studied comments carefully and have made correction which we hope meet with your approval. Revised portions are marked in red in the paper. The main corrections in the paper and the responds to the comments are as flowing.
Responds to the reviewer’s comments:
1.Response to comment: Title - you can change glacier no. 1 to something more general like "a glacier", or you can change the title to: Bacterial diversity analysis and screening for ACC deaminase-producing strains in moss-covered soil at different altitudes in Tianshan Mountains - a case study of Glacier No. 1"
Response: change the title to: Bacterial diversity analysis and screening for ACC deaminase-producing strains in moss-covered soil at different altitudes in Tianshan Mountains - a case study of Glacier No. 1" (line2-5).
2.Response to comment: All the text - pay attention to Latin names of genus and species to be written in italic (e.g. line 49).
Response: The Latin names of genera and species have been rechecked for italicization throughout the text (e.g. line 29, line336-338).
3.Response to comment: Keywords - change the keywords that are present in the Title with other similar ones. In this way, you can expand the list of words/ concepts related to your work.
Response: Keywords changed to: Global warming, Primary succession, High-throughput sequencing, ACC deaminase (line34).
4.Response to comment: Introduction - in the first paragraph, more international references to sustain the problem of climate changes and their effect on glaciers. Also, you can add one or two sentences to describe the glaciers as unique ecosystems. This is vital for the importance of such type of studies.
Response: This section has been rewritten according to the revision, with reference to relevant national and international studies describing the issue of climate change and its impact on glaciers, as well as adding the description of glaciers as unique ecosystems (line37-48).
5.Response to comment: Try to use grassland instead of meadow in the text.
Response: Grassland has been replaced with meadow in the text (line59).
6.Response to comment: The last paragraph of the introduction should present a clear aim of the research, with separate sentences for objectives/hypotheses proposed by the authors.
Response: The last paragraph of the introduction to this article has been re-edited (line85-90).
7.Response to comment: It would be interesting to make a sampling scheme where to point visually the sampling procedure. This will help other researchers to replicate this type of research.
Response: Split the 2.1 sampling section of the original text into two parts: study area overview and sampling scheme (line92-111).
8.Response to comment: Add the reference for Manual of Systematic Identification of Common Bacteria and Microbiological Experiments - lines 156-157.
Response: Modified according to expert recommendations (line151-152).
9.Response to comment: Add a data analysis subsection to explain which type of test are used, and why, to analyze the results. Also, the software is important to be presented and referenced.
Response: Add subsection Statistical Analyses (line164-169).
10.Response to comment: Expand the interpretation of Table 2 and use the words significant (where is the case) to analyze the differences between sampled location.
Response: Modified according to expert recommendations (line175-180).
11.Response to comment: Expand the interpretation of Table 3 and Figure 1. You have interesting results, they need to be fully presented.
Response: Modified according to expert recommendations (line203-204, 206-209).
12.Response to comment: Table 4 - similar suggestion as above.
Response: Modified according to expert recommendations (line253-261).
13.Response to comment: Figure 3 -the variance explained by PCA and RDA is low. Expand the interpretation and analyze if the low variance o not affect the quality of results presentation. It can be used other ordination as NMDS or PCoA if they present clearer the results. Or, try to make another PCA and RDA, additional to the ones already present where to exclude AY samples. Figure 3 will have 4 graphs, and you can discuss on the differences between communities and within communities. I suggest the last solution. This will add more results to be presented.
Response: Modified according to expert recommendations (line267-279).
14.Response to comment: Subsection 3.6. Figure 5 do not give much information to increase the quality of the study. It can be removed.
Response: Deleted Subsection 3.6 Figure 5.
15.Response to comment: Expand the interpretation of Figure 6. Here you need to point with more explanations why did you choose the three strains to further test.
Response: Modified according to expert recommendations (line315-320).
16.Response to comment: Expand the interpretation of Biolog identification results (3.7.3.). Analyze the different use of substrates and the specific patterns for each. This will help you to complete the Discussion section. Connect the 3.7.3 and 3.7.4. subsections.
Response: Modified according to expert recommendations (line340-350).
17.Response to comment: First paragraph is more useful in the Introduction (Lines 346-351) it does not help in this section.
Response: The first paragraph of the discussion has been deleted.
18.Response to comment: Rewrite the sentences from 456-463 to exclude the word such speculation. Try to identify references that sustain this affirmation or to propose a new explanation. Use words such "a possible explanation", "this phenomenon was caused by" or something similar.
Response: Modified according to expert recommendations (line470-471).
Special thanks to you for your good comments.
Sincerely,
Corresponding author:
Name: Yonghong Fan E-mail: fyh@xju.edu.cn

Round 2
Reviewer 2 Report
Dear authors,
The new form of the manuscript presents better the results. There is a minor aspect that need to be modified in order to present in a clearer manner the aim and the objectives. Split the last paragraph of the Introduction section in short sentences. One for the aim and separate sentences ofr objectives.
The manuscript is interesting and present a nice flow of the research.